# Digital Motion Graphics and Animated Media in Health Communication: A Systematic Review of Strategies for Sexual Health Messaging

**DOI:** 10.3390/healthcare13222895

**Published:** 2025-11-13

**Authors:** Nattawat Suwanphan, Dichitchai Mettarikanon, Siriwatchana Kaeophanuek, Chime Eden, Weeratian Tawanwongsri

**Affiliations:** 1School of Informatics, Walailak University, Nakhon Si Thammarat 80160, Thailand; nattawat.suw@wu.ac.th (N.S.);; 2Informatics Innovation Center of Excellence, School of Informatics, Walailak University, Nakhon Si Thammarat 80160, Thailand; 3Division of Dermatology, Jigme Dorji Wangchuck National Referral Hospital (JDWNRH), Thimphu 11001, Bhutan; 4Division of Dermatology, Department of Internal Medicine, School of Medicine, Walailak University, Nakhon Si Thammarat 80160, Thailand

**Keywords:** motion graphics, animation, digital media, health communication, public health messaging, sexual health education

## Abstract

**Background**: The influence of digital media on public perception of sexual health is significant and relates to its effects on knowledge, attitudes, and behaviors. Motion graphics and animation represent a novel innovation in health-related communication. The visual attractiveness and adaptability to cultural issues offer an alternative means of conveying often sensitive issues. This systematic review aimed to provide an overview of their efficacy in delivering sexual health messages and to identify the components contributing to success. **Methods**: A literature search was conducted using Scopus, MEDLINE (via PubMed), and DOAJ, covering studies from inception to 31 August 2025. All eligible studies included randomized controlled trials, quasi-experimental studies, and other evaluations, which were synthesized by outcomes related to knowledge, attitudes, behaviors, intentions, and/or psychosocial well-being. The study was pre-registered on the INPLASY platform (INPLASY202580073). **Results**: Eleven studies published between the years 1989 and 2025 met the inclusion criteria. The majority reported improvements in sexual health knowledge and attitudes, and several studies also demonstrated improved behaviors and psychosocial well-being. Factors that contributed to interventions that have been successful were cultural adaptation, being based on theory, and presentation over time. **Conclusions**: Motion graphics and animation, therefore, represent an exciting means of sexual health promotion and stigma reduction. Future studies should therefore focus on the determination of standardized formats for media presentations, the evaluation of long-term effects, and the evaluation of cost–benefit to enhance the effectiveness of media communication in health promotion.

## 1. Introduction

Sexual health education is a comprehensive approach that is biopsychosocial and promotes the understanding of biological, psychological, social, and reproductive health. The goal of sexual health education is to provide patients with the necessary knowledge, skills, and positive attitudes to make decisions to prevent adverse health outcomes such as sexually transmitted infections (STIs) and unintended pregnancies and promote autonomy in sexual health [1,2,3]. Sexual health education improves health outcomes. Increases in knowledge and understanding promote protective behaviors in which individuals engage in consistent contraceptive and barrier use, thus decreasing the rates of risky sexual behaviors among adolescents [4,5,6]. These benefits are then reflected in the lower rates of STIs (including HIV) and fewer unintended pregnancies [7]. Sexual health education also contributes to psychosocial health by fostering respect, communication, and consent in relationships—protective factors against sexual violence and sexual abuse [8,9]. Furthermore, inclusive curricula promote the acceptance of diversity and stigma reduction and are especially beneficial for lesbian, gay, bisexual, transgender, queer/questioning, and other sexual and gender minority (LGBTQ+) youth [10]. The benefits of sexual health education go beyond individuals and affect educational and social outcomes. Schools with sexual health education can better engage their students academically, who may be faced with fewer distractions related to sexual health or a more inclusive climate [11]. As beneficial as sexual health education is, barriers such as limited funding, limited teacher training, and socio-cultural resistance hinder the delivery of sexual health education worldwide [12].

Considering these challenges, innovative educational media in the form of motion graphics and animated videos may be valuable tools for effectively enhancing the delivery and engagement of sexual health education. Animated video materials have been shown to considerably increase student engagement and interest in various contexts. For example, animated videos lead to improved understanding and self-directed learning among undergraduate accounting students [12], as well as increased motivation and acquisition of motor skills in a physical education setting [13]. Animated videos are not only beneficial for student engagement but also for improved learning outcomes. Students who learned vocabulary or psychology content through animations scored higher on tests and preferred animated videos over traditional materials [14,15]. Additionally, animated videos can improve the comprehension of complex concepts found in areas such as machine learning and natural sciences, as there is a comprehension benefit from the visualization and memorization of abstract learning content [16].

Motion graphics are a screen-based medium that creates the illusion of movement by sequencing images, often combined with text, audio, and numerous multimedia factors. As a hybrid visual communication, they provide additional clarity, engagement, and emotional depth [17]. Instead of a narrative focus, as seen in animations, motion graphics use animated graphic elements with low-stylized representations to convey information and concepts [18]. Their uses range from advertising to education and from design interfaces to digital content, and this distinct visual storytelling technique has become increasingly popular as a vehicle for communicating information and concepts [19]. Like motion graphics, animated videos are audiovisual productions that communicate information, ideas, and narratives through movement using animation approaches, such as hand-drawn illustrations, computer-generated imagery, or frame-by-frame production. Animated videos combine vibrant visual forms (such as drawings, models, or images) with audio elements (such as sound effects, narration, and music), not only to motivate learners but also to provide an interactive effect to aid understanding [20,21]. Unlike motion graphics, animated videos creatively portray concrete realities and abstract concepts, the latter of which can depict events or concepts that cannot be easily captured using video and storytelling. Animated documentaries have depicted psychological experiences, events from the past, or art that has never been recorded in audiovisual form, reestablishing the referential value of digital media.

While sexual health education has seen widespread advancement in practice, it is typically designed using conventional text-based modes of delivery or traditionally didactic approaches that do not engage learners effectively or consider sensitive topics in accessible ways. Recent evidence shows that motion graphics and animated videos can improve engagement, understanding, and retention when implemented in various educational modalities; however, the utilization of these innovations in sexual health education has yet to be systematically summarized. Individual studies have reported positive improvements in sexual health knowledge, attitudes, and protective behaviors, and even decreased risky behaviors in sexual health when using motion graphics/animated videos. However, these interventions also varied in design domains, populations, theories informing the designs, and levels of quality, making it important to conduct a review in this area. A review summarizing the current state of knowledge with respect to interventions provides information on consensus and gaps, and assists in informing future intervention designs. This systematic review examines the use of motion graphics and animated videos in sexual health education. More specifically, the review will (i) describe the features and design characteristics of these interventions; (ii) describe the educational and behavioral outcomes addressed with motion graphics and animated videos in sexual health education; and (iii) summarize the evidence of the effectiveness of motion graphics and animated videos in improving knowledge, attitudes, behaviors, and related outcomes in sexual health.

## 2. Materials and Methods

This systematic review adheres to PRISMA 2020 guidelines for systematic reviews [22] (Appendix A). The review protocol was prospectively registered using the INPLASY International Platform of Registered Systematic Review and Meta-analysis Protocols (INPLASY registration number: INPLASY202580073). This study was approved by the Walailak University Ethics Committee (WUEC-25-326-01).

### 2.1. Eligibility Criteria

The eligibility and inclusion criteria were determined a priori. Studies were eligible if they examined a sample of any population regardless of age, sex, or setting and assessed an intervention addressing sexual or sexual health content that used motion graphics, animated videos, educational animations, digital animation, explainer videos, 2D/3D animations, or animated infographics. There were no restrictions on comparators (e.g., standard education, some other intervention, or no intervention). To be included in the review, studies were required to assess and report on at least one relevant outcome related to sexual health, specifically knowledge, awareness, attitudes, behaviors, behavioral changes, intention, uptake, or practice. Both experimental (e.g., randomized controlled trials, quasi-experimental, or pre–post studies) and other quantitative evaluation designs (e.g., observational or pilot interventions reporting measurable outcomes) were eligible for inclusion. Only the studies published in English were eligible for inclusion. If researchers could not retrieve the full text of a study through reasonable means, it was dropped from the inclusion criteria. The eligible studies were later grouped for synthesis based on intervention characteristics and reported outcome measures.

### 2.2. Information Sources

A literature search was performed using three major electronic databases: Scopus, MEDLINE (via PubMed), and Directory of Open Access Journals (DOAJ). The databases were searched from their inception to 31 August 2025. The reference lists of included articles were screened for additional relevant studies. The grey literature and unpublished studies were not searched in this review, as the aim was to synthesize evidence from the peer-reviewed published literature to ensure a high-quality method, reliability, and replicable study findings.

### 2.3. Search Strategy

A comprehensive search strategy was developed for each database, in conjunction with controlled vocabulary terms and free-text keywords. The search examined Scopus, MEDLINE (via PubMed), and DOAJ, from inception until 31 August 2025, using filters for English-language publications. The search strategy was a combination of terms related to the intervention (e.g., motion graphics, animation) and the topic (e.g., sexual health, HIV prevention, sex education), along with relevant outcomes (e.g., knowledge, attitudes, behaviors, and intention). The complete search strategy for Scopus is outlined below as an example, with similar strategies adapted for other databases.

TITLE-ABS-KEY(

(“motion graphic*” OR animation* OR “animated video*” OR “educational animation*”

OR “explainer video*” OR “digital animation*” OR “2D animation”

OR “3D animation” OR “animated infographic*”)

AND

(“sexual health” OR “sex education” OR “HIV prevention” OR “HIV education”

OR “HIV testing” OR condom* OR PrEP OR “pre-exposure prophylaxis” OR PEP

OR “post-exposure prophylaxis”)

AND

(knowledge OR awareness OR attitud* OR behav* OR “behavioral change”

OR intention* OR uptake OR practice*)

)

AND (LIMIT-TO (LANGUAGE, “English”))

### 2.4. Selection Process

All records identified from the database searches were imported into EndNote version 20 (Clarivate Analytics) and duplicates were removed. Title and abstract screening was conducted independently by two reviewers (NS and WT) according to the eligibility criteria stated above. The full texts of the potentially relevant articles were subsequently retrieved and independently assessed by the same reviewers. We consistently applied our eligibility criteria across both screening stages. Any conflicts regarding study selection were resolved through discussion, and if they could not be resolved, a third reviewer (DM) was consulted. No automation tools were used in the selection process.

### 2.5. Data Extraction and Synthesis

Two reviewers (NS and WT) independently extracted study data using a standardized extraction form. For each included publication, the data extracted included the authors and year of publication, country of the study, specific focus of the sex education topic, user characteristics, study design, details of the intervention, measures, and study findings. Educational and multimedia details of the motion graphics and animated video interventions were also extracted, including the type of animation (e.g., 2D, 3D, or mixed), duration and frequency, content focus (e.g., abstinence and contraception), delivery platform (e.g., mobile app and online video), pedagogical/educational design features (e.g., quizzes and gamification strategies), level of user interactivity, and any cultural or contextual adaptation described. Production and expertise information (i.e., who created or recommended the material) was also recorded. Data discrepancies were resolved through a discussion process involving all reviewers, and adjudication was sought from a third reviewer (DM) when reviewers could not come to a consensus. No automation tools were used, nor did we seek additional data from study authors.

Due to a marked level of methodological, clinical, and conceptual heterogeneity (across populations, settings, intervention formats, comparators, outcomes, and lengths of follow-up), a quantitative meta-analysis was not possible. Instead, we produced a narrative synthesis. Studies were first defined and grouped by the sexual health domain (knowledge, attitudes, behavioral intention, and psychosocial outcomes) and then by study design and population group (e.g., adolescents in school, university-based students, and persons living with HIV). Within each group, we compared the direction and statistical significance of effects, reported estimates of effect (when possible), and consistent effects across studies. Any contradictory or outlying effects were reported rather than statistically pooled. This strategy allowed us to report patterns of effects while concurrently accounting for heterogeneity in designs and outcomes.

### 2.6. Bias Assessment

The methodological quality of the included studies was evaluated according to the study design. For RCTs, the potential risk of bias was assessed using the Cochrane Risk of Bias 2 (RoB 2) tool [23], which assesses the potential risk of bias in the RCT design according to five domains: randomization process, deviations from the intended intervention, missing outcome data, measurement of the outcome, and selection of the reported result. For non-randomized studies, the potential risk of bias was assessed using the Risk Of Bias In Non-randomized Studies of Interventions (ROBINS-I) tool, which has seven domains: confounding, selection of participants into the study, classification of interventions, deviation from the intended intervention, missing data, measurement of outcomes, and selection of reported results [24]. The risk of bias assessments were independently performed by two reviewers (NS and WT). Disagreements were resolved by discussion, and a third reviewer (DM) was consulted whenever necessary. Risk-of-bias judgments were subsequently taken into account when interpreting the direction and magnitude of effects within each outcome domain and when describing the overall strength of evidence in the Results and Discussion.

### 2.7. Assessment of the Strength of Evidence

Due to the limited number of studies in particular outcome domains and the high degree of design, intervention, and outcome measure heterogeneity, we did not conduct a formal Grading of Recommendations Assessment, Development and Evaluation (GRADE) rating for each outcome. Instead, we used a qualitative approach to assess evidence strength for each outcome domain (knowledge, attitudes, behavioral intention, and psychosocial well-being) by considering (i) study design (i.e., RCT versus non-randomized), (ii) overall risk-of-bias profile using RoB 2 and ROBINS-I, (iii) direction of effect (if any) was consistent, (iv) indirect nature of populations, intervention, and outcomes, and (v) precision of the reported estimates (e.g., confidence intervals, *p*-value, effect size if available). Given this approach, we have summarized the evidence as stronger or weaker in the Results and Discussion and not used formal categorical certainty ratings.

## 3. Results

Eleven studies published between 1989 and 2025 were included in this review (Table 1 and Table 2). Figure 1 illustrates the overall process flow for study selection in a PRISMA flow diagram. The studies were conducted in China (n = 5) [25,26,27,28,29], the United States (n = 3) [30,31,32], Malaysia (n = 1) [33], Tanzania (n = 1) [34], and Thailand (n = 1) [35]. The risk of bias assessment for the included studies is summarized in Table 3 and Table 4.

Interventions addressed a range of sexual health topics; that is, HIV or STI prevention (n = 5) [27,28,31,34,35], sexting prevention (n = 1) [33]; HPV vaccination intention (n = 1) [26], sexuality education more broadly (n = 2) [25,29], pregnancy prevention (n = 1) [32] or adherence/medication-related knowledge (n = 1) [30].

The methodological designs were diverse: RCTs (n = 4) [25,32,33,35], quasi-experimental studies (n = 2) [27,28], longitudinal community-based studies (n = 1) [34], and pre–post or pilot evaluations (n = 4) [26,29,30,31]. The sample size varied from 51 participants in a U.S. university study [30] to 1835 preadolescents in a large-scale school-based intervention in China [29].

The study population was heterogeneous. These included preadolescents aged 9–12 years in rural or primary schools (n = 2) [28,29], adolescents in vocational or secondary schools (n = 2) [25,34], diploma-level university students (n = 1) [33], high school students who presented to clinics (n = 1) [32], foreign-born Latino men (n = 1) [31], HIV-positive men who have sex with men (n = 1) [27], HIV-infected adolescents and their caregivers (n = 1) [35], and adult university/community populations (n = 2) [30,34].

### 3.1. Intervention Characteristics

Among the eleven studies included in this review, the most common delivery method for animated and motion-graphic interventions was face-to-face in classrooms (n = 5) [25,28,29,32,34]. These interventions were typically delivered through animated videos and structured lessons led by teachers and included interactive components such as discussions and quizzes. The second most common delivery method was digital or online (n = 4) [26,27,30,33], in which animated videos were provided via online surveys, private YouTube links, interactive games, or handheld devices. Two studies were conducted in clinical or community settings, using cartoon-style animated videos within family sessions in HIV clinics or individually tailored animated modules delivered on tablets [31,35].

The duration and frequency of interventions varied across studies. Three studies used brief single-session interventions [26,30,31], whereas four studies implemented multi-session curricula lasting several weeks [25,28,29,32]. Four studies reported longer-term interventions delivered over several months, with follow-up extending to one year [27,33,34,35].

Most interventions (n = 8) incorporated interactive components such as group discussions, quizzes, individualized feedback, or follow-up conversations [25,27,28,29,32,33,34,35]. Regarding animation modality, two studies explicitly reported using 2D animation [25,35]. None explicitly reported 3D or mixed formats; the remaining studies did not specify the animation type. Cultural or contextual adaptation was noted in nine studies, most often as surface adaptations (e.g., translation or local iconography) and, in several cases, as deep-structure adaptations involving community or stakeholder input [25,26,27,28,29,31,32,33,35]. These characteristics are summarized in Table 2.

### 3.2. Theoretical Frameworks and Pedagogical Approaches

Eight of the 11 studies included in the review clearly used theoretical or pedagogical frameworks to inform the design of their interventions. The Prototype Willingness Model (PWM) was used in one study [33] to address sexting behaviors, whereas Li et al. [26] applied the Behavioural and Social Drivers (BeSD) framework to assess willingness for HPV vaccination. The CHAMP+ Thailand family intervention was based on Social Action Theory [35], and Paperny and Starn [32] used Social Learning Theory in their interactive pregnancy-prevention games.

Several school-based programs were guided by comprehensive sexuality education standards from UNESCO or UNFPA, aligned with local engagement and social norms theory [25,28,29]. Brock and Smith [30] used self-efficacy theory to improve adherence to antiretroviral therapy in the United States, and Grieb et al. [31] combined the Information–Motivation–Behavioral Skills (IMB) model with the Transtheoretical Model to promote HIV testing. Three studies did not cite a guiding theoretical model but employed pedagogical strategies such as peer education, co-design with local actors, or integration of digital health literacy principles [27,29,34].

In terms of pedagogy, most interventions emphasized interactivity (n = 8), including quizzes, discussions, storytelling, or follow-up activities. Storytelling was most often employed to enhance comprehension among low-literacy participants or to strengthen cultural connectedness, as seen in CHAMP+ Thailand [35] and the co-created HIV testing module for Latino men [31]. Cultural tailoring and linguistic adaptation were incorporated in nearly all studies to maximize acceptability.

### 3.3. Educational Outcomes

All 11 studies reported sexual health education-related outcomes, with the majority reporting knowledge (n = 9) and attitudes (n = 7), and some assessing intentions or behavioral outcomes (n = 6). Overall, the studies indicated that, in general, motion graphic and animated video health interventions increased individual sexual health knowledge, improved attitudes toward safer sex practices and inclusivity, and, in some cases, impacted intentions and behaviors related to sexual health prevention and decision-making.

Knowledge-related outcomes were reported in nine studies [25,26,27,28,29,30,31,34,35]. Improvements in knowledge were widely documented across the studies, ranging from directed short-term gains resulting from a brief HPV vaccination video [26] to short-term statistically significant acute increases in understanding of HIV medication after viewing a 17 min digital video among U.S. university students, with effects sustained over a 4–6 week period [30]. The sustained effects of sexual health programs and positive changes in sexual knowledge have also been seen in large school-based programs, with scores on sexual knowledge tests improving by 3.94 points immediately and 4.38 points at 1 year [29] and a sustained increase in Tanzanian health education programs with 10–31% higher knowledge about HIV/AIDS and parasitic infections at 1 year [34]

Attitudinal outcomes were explored in seven studies [25,28,29,31,32,33,35]. These studies reported positive attitudes such as greater inclusion of LGBTQ identities [25], decreased stigma and internalized shame among HIV-infected youth [35], increased contraceptive attitudes in U.S. adolescents [32], and greater openness around HIV testing by Latino immigrant men [31]. In larger school-based interventions, attitudinal changes were observed that decreased slightly over time [29].

Intentions and behavioral outcomes were assessed in six studies [26,27,29,31,32,33]. Mansor et al. (2023) [33] reported significant reductions in intention and willingness to sext. Li et al. (2025) [26] reported an increased willingness to receive HPV vaccinations. Grieb et al. (2017) [31] reported an increase in HIV testing among Latino men, and Mi et al. (2015) [27] reported improved HIV diagnosis disclosure and uptake of partner testing among MSM. Paperny and Starn (1989) [32] noted an overall decrease in adolescent pregnancies in clinics that utilized their computer-assisted games.

### 3.4. Behavioral and Psychosocial Outcomes

Six studies reported outcomes that extend from knowledge and attitudes to the behavioral and psychosocial domains [26,27,29,32,33,35]. There were several behavioral outcomes. Mansor et al. (2023) [33] indicated that among Malaysian diploma students who engaged in a short-animated video module, there were significant reductions in intention to sext (β = −0.12, *p* = 0.002) and willingness to sext (β = −0.16, *p* < 0.001). Li et al. (2025) [26] reported that 88% of Chinese college students indicated a willingness to receive HPV vaccination after viewing a two-minute HPV prevention video. Mi et al. (2015) [27] found that HIV-positive MSM who engaged in a web-based program with an animated game were more likely to disclose their HIV status to their partners (76% vs. 61%) and encourage their partners to undergo testing (42% vs. 26%). There was evidence of behavioral outcomes at a community level as well. Paperny and Starn (1989) [32] reported that clinics implementing computer-assisted games recorded a 15% reduction in one year of adolescent positive pregnancy tests.

Three psychosocial outcomes were examined [29,34,35]. Nestadt and colleagues found that the CHAMP+ Thailand family-based cartoon curricular approach led to significant improvement in youth mental health, HIV knowledge, adherence, caregiver–child communication, and reduced internalized stigma at 6 months post-intervention that were mostly sustained at 9 months. Zhou et al. (2024) [29] did not find a significant change in sexual attitudes immediately or at one year post-intervention, but there did appear to be improvements, and any change in attitudes degraded over time, indicating that significant attitude change in large-scale school interventions is unlikely to be sustained over time. Holst and colleagues (2022) [34] described psychosocial gains in Tanzania where youth who accessed community “InfoSpots” between animated health education reported increased engagement with health information and reinforced learning over the following 12 months.

### 3.5. Summary of Intervention Outcomes

A summary of findings across the 11 studies—covering knowledge, attitudes, behavioral intention, and psychosocial well-being—is presented in Table 5.

## 4. Discussion

The results of this review show that motion graphics and animated videos have the potential to serve as new tools for sexual health education. Across the studies included in this review, the media consistently supported knowledge gain, attitude improvement, and intentions, behavior, and psychosocial outcomes in numerous studies. Interestingly, the effects were observed across a variety of populations, from preadolescents in school to HIV-positive adults in clinical care, and in some studies lasted up to one year after the intervention ended. Overall, these findings demonstrate the versatility of animated interventions across contexts and illustrate the ability of animated interventions to get participants to be more engaged than standard didactic teaching.

### 4.1. Motion Graphics and Animated Video Interventions and Educational Outcomes

The consistent knowledge gains observed across diverse populations support the idea that motion graphics and animated videos allow and advance knowledge processing of sexual health information because the content is delivered in an engaging multimedia format that simultaneously signals two channels for learning and application. Sustained knowledge improvements in school-based programs through social norms [28,29] suggest that delaying repeat exposures’ effects, with the potential of teacher-facilitated delivery, may allow knowledge to be reinforced and retained more than if just once, as noted with brief interventions, such as short HPV or HIV prevention videos, while the studies reported show short-term [26,30] knowledge gain and that well-designed, brief, low-dose learning interventions could be impactful. Attitudinal outcomes also suggested a positive change in terms of greater inclusivity toward LGBTQ+ identities [25,35] and more information and less stigma toward topics of sexual health and wellbeing. These studies demonstrate that animation is a safe place to examine sensitive or value-laden content. A decrease in attitudinal improvements over time suggests that in the absence of reinforcement, attitudinal change may be less stable or durable than knowledge outcomes [29]. An important variation in the studies reviewed is the durability of the effects. Although brief, one-time interventions such as short HPV or HIV videos showed immediate improvements in knowledge [26,30], indicating that animation is an effective way to engage learners quickly and get the key messages across. There was no question whether multi-session curricula designed to consider their placement within school programs had both immediate and sustained improvements in knowledge and, to a lesser extent, in attitudes up to one year post-delivery [29,34]. This indicates that short interventions can be efficient and scalable, but longer or more consistent exposure may be required to improve learning and change attitudes. The decline in attitudes over time in some larger-scale programs underlines the importance of thinking about reinforcement strategies to maintain psychosocial impact [29].

With respect to promoting positive learning outcomes, it can be construed that several theoretical constructs might elucidate why motion graphics and animated video interventions have an effect. Among the more broadly articulated constructs is dual coding theory, as discussed by Arora and Giri (2022) [36], which asserts that information is processed through both verbal and non-verbal systems. Each information system can be encoded to provide an equal yet distinct representation of conceptual information. If the content is delivered via both dual channels simultaneously or close in time, learners can build information with richer mental representations that will ultimately lead to a significantly deeper understanding and improved recall [37]. This principle appears to be echoed in research related to sexual health education, where narrated animations appear to increase the engagement and retention of knowledge [29]. Also, similarly aligned with multimedia learning research, the perspective of lessening cognitive load in relation to animations implies that both visual and verbal inputs may increase comprehensibility of material [38,39]. Collectively, these theories would support the premise that positive learning outcomes cited in the studies across the previous studies were tied to the nature of dual-channel processing.

Instructional animations are generally more interesting and more likely to attract learners’ attention than traditional text-based approaches. While text conveys information through printed words, animation uses a combination of visual cues, character design, voice acting, and dialogue to create a more enjoyable, motivational learning experience that also contributes to comprehension and retention [40]. In particular, when combining narratives with animated characters, learners enjoy the experience more and create an anxiety-free atmosphere in the learning environment, allowing for better separation between themselves and the difficult content [41]. It has also been demonstrated empirically that motion graphics can increase engagement and positively correlate with understanding, while alleviating stressors sometimes associated with traditional printed materials. These characteristics further demonstrate the role of engagement as a salient mechanism—both as a mediating variable influencing learning outcomes and as a protective factor that mitigates cognitive load.

While evidence of consistent positive change across studies has been observed, there is a need to interpret the level of evidence cautiously. Several studies included, and many non-randomized studies, were rated with a serious or moderate risk of bias primarily due to confounding, selection of participants, or measurement of outcomes. These biases could be overestimating the intervention or creating uncertainty about the generalizability of findings. There was too much methodological heterogeneity and variability in the quality of studies to do any quantitative weighting or sensitivity analysis. Thus, the studies and findings included in this review reported a qualitative synthesis, and future studies should utilize more rigorous designs, standardized outcomes, and transparently report quality to support the reliability of that evidence.

### 4.2. Motion Graphics and Animated Video Interventions and Behavioral and Psychosocial Outcomes

In addition to knowledge and attitudinal gains, some interventions showed substantial differences in behavioral and psychosocial outcomes. These animated modules, which specifically targeted sexting behaviors among Malaysian diploma students, reported reduced sexting intention and willingness. These results suggest that theory-based intervention designs can conceptualize ways to modify risky digital behavior [33]. Similarly, a brief HPV video intervention increased the likelihood of vaccination intention among Chinese college students, indicating that even brief exposure to animated and digitized health education can help positively frame health prevention intentions [26]. Behavioral change was also indicated in HIV-related contexts, where personalized animated videos increased HIV testing intention with Latino immigrant men [31], and a web-based program that included an animated game promoted the disclosure of HIV status and partner testing with Chinese MSM [27]. Psychosocial gains were noted in family-based interventions in which cartoons were used as part of a curriculum to improve communication, decrease internalized stigma, and promote adherence among Thai adolescents living with HIV [35]. Community-based delivery models may also have implications for learning and engagement; Tanzanian participants reported increased knowledge gains and continued use of digital InfoSpots to seek health information [34]. Overall, these findings suggest that animated interventions can positively impact not only knowledge and awareness, but also actual behaviors and psychosocial well-being across a diverse range of cultural and clinical situations.

While multiple interventions showed short-term behavioral outcomes, many did not show sustained behavioral changes. Reductions in sexting intention and willingness among Malaysian diploma students were sustained three months after exposure to animated prevention modules, highlighting how effective repeated interventions based on theory can achieve enduring changes in risky digital behavior beyond the short term [33]. In one study on HIV prevention, researchers reported evidence of sustained explicit disclosure of HIV status and partner testing encouragement among Chinese MSM six months after participation in a web-based program that included an animated game. The authors suggested that animation was linked to the durability of behavioral change, especially when contained within multi-component web-based interventions [27]. Similarly, the CHAMP+ Thailand program improved adherence and family communication and reduced internalized stigma at nine months among HIV-infected adolescents [35]. These studies support the observation that animation may produce behavioral intention changes and sustainable behavioral changes when delivered in combination with curricula, peer support mechanisms, or family support mechanisms.

The identified changes in intentions, behaviors, and psychosocial well-being can be further contextualized using established behavioral change frameworks. The four models most relevant to the interventions reviewed are the PWM, Social Cognitive Theory (SCT), the Health Belief Model (HBM), and the IMB Model. Each model offers important features that explain the ability of animated interventions to impact decision-making, encourage protective behaviors, and address stigma.

PWM often views health behavior as a reasoned process of intentional behavior. In contrast, PWM delineates two separate pathways to health behavior: a reasoned pathway driven by deliberate intentions and a pathway driven by social reaction that involves the willingness to engage in behavior and heuristic processing [42]. The potentially unique constructs of PWM are behavioral willingness—an individual’s openness to engaging in a behavior whenever they are presented with an opportunity to do so—and prototypes, images of the typical person who engages in the behavior. These social images can have a particularly strong influence on adolescents, and importantly, adolescent willingness often predicts risky behavior better than intention at a younger age. Importantly, prototype favorability and similarity are not fixed and may be actively shifted by interventions. Evidence of these principles of PWM was confirmed in animations intended to effectively create new prototype perceptions and social norms: decreased willingness to sext in sexting prevention modules for Malaysian diploma students [33] and increased willingness to vaccinate against HPV following a brief HPV-related video for Chinese university students [26]. The use of animation to create attitudinal change demonstrates how animated intervention can actualize PWM by changing young people’s social images and limiting their reactive engagement in risky behaviors.

SCT elucidates human behavior as a function of the interconnectedness of three dynamic and reciprocal factors: personal, behavioral, and environmental influencers. The schematic provides a framework for individuals’ self-efficacy and outcome expectancies related to personal behavior and notes the socio-structural influences on behavior, which are the central constructs of SCT [43]. In educational contexts and health promotion, animations serve to model appropriate behaviors, mitigate stigma, and enhance self-efficacy; to accomplish these educational goals, animations have relatable characters and scenarios, as illustrated in the CHAMP+ Thailand program that used cartoons in sessions to enhance communication, improve adherence to treatment, and reduce stigma among HIV-infected adolescents and their caregivers [35]. In the program, the intervention highlighted the coping strategies and provided role models leveraging observational learning, a key SCT principle intended to normalize conversations while reinforcing psychosocial behaviors to respond adaptively. Overall, this work illustrates how animations are an opportunity to put SCT principles into practice and foster individual capacity for agency, as well as build supportive social contexts for individuals to follow through with sustained altered behavior.

HBM conceptualizes preventive health behavior as being shaped by individuals’ perceptions of susceptibility, severity, benefits, barriers, and cues to action [44]. Animated interventions can increase perceived susceptibility and perceived benefits while reducing perceived barriers through clear and relatable messaging. For example, a two-minute HPV vaccination video increased the willingness for vaccination among Chinese university students [26], whereas animated educational modules targeting barriers to HIV testing increased testing intentions among Latino immigrant men [31]. In both these examples, the animation functioned as a cue to action by making abstract risks more salient and bringing the benefits of preventive behaviors to the surface. Additionally, the HBM is applicable across diverse groups, which supports its utility in explaining why short, visually oriented media can encourage individuals to transition from awareness to action for prevention [45].

Finally, the IMB model argues that three elements—accurate information, sufficient personal and social motivation, and skills needed to enact a behavior—must occur together to accomplish health behavioral changes [46]. Tailored animated interventions can potentially meet all three. The tailored video modules, individually focused on HIV prevention and care designed for Latino immigrant men and increased HIV testing intentions by providing information about testing, addressing motivational barriers such as stigma, and including skill-building content to access local services [31]. This demonstrates the strength of the model: animations can combine knowledge with motivational cues and problem-solving strategies that strengthen self-efficacy and reduce obstacles to action. Although it was initially designed for HIV-related behaviors, the organized yet adaptable nature of the IMB framework makes it particularly appropriate for multimedia interventions that blend storytelling, visual prompts, and culturally appropriate messaging [47].

### 4.3. Conceptual Synthesis and Unified Framework

We synthesized the behavioral and learning theories referenced in the included studies and the literature, which encompass HBM [44], PMT, TRA/TPB [42], SCT [43], IMB [46], cognitive-affective learning theories (dual coding and coherence/signaling models), and narrative persuasion theory [36,39]. Figure 2 provides an overview of a model for some potential ways that animated and motion-graphic interventions might influence behavior change. In this model, design components (e.g., story, relatable characters, signaling, pacing, dual audio-visual cues, interactivity) and delivery features (e.g., setting, dosage/repetition, fidelity, access) influence proximal cognitive-affective mediators, including attention and engagement, comprehension and memory, emotion and identification, and perceived norms. These mediators in turn shape beliefs and efficacy (e.g., susceptibility, severity, benefits, barriers, self/response efficacy) factors (i.e., HBM, PMT, SCT, IMB), which influence attitude and intention (i.e., TRA/TPB) factors or constructs. These cognitive processes result in behavior (e.g., HIV testing, vaccination, condom use) as well as the sustaining of prevention behaviors.

The model also accounts for moderating factors—health & digital literacy, gender & diversity, prior knowledge, stigma, digital access/bandwidth—as well as variables about the quality of the intervention (i.e., dose, fidelity, access) that may increase or decrease the overall efficacy of the intervention. This conceptual model provides clarity on where animated media will likely exert the most influence on behavior measures, through attention, comprehension, and identification with narrative, and demonstrates where future research will need to begin developing measurement instruments to assess these mediators and moderators in addition to measuring for equity-sensitive outcomes [39,42,46].

### 4.4. Health and Digital Literacy as Moderators of Impact

The impact of animated or motion graphics interventions is likely to be modified by learners’ health and digital literacy. For those possessing lower health literacy, it is more difficult to understand, remember, and act on health information, which may reduce the impact of otherwise appealing media. Recent prospective data demonstrate that ineffective health literacy is associated with poor outcomes and greater health service usage, which suggests the importance of designing for low-literacy audiences and measuring literacy in subsequent trials [48]. Programs should, therefore, employ HL-informed design (simple language and concise messages, captions/voice-over in local language, and high contrast/icon-oriented visuals) and evaluate using brief teach-back prompts or facilitation.

### 4.5. Equity Considerations: Gender, Diversity, and Health/Digital Literacy

The efficacy and the equitability of animated sexual-health messages will likely differ by gender, sociocultural background, and health/digital literacy. Certainly, there is evidence of gender differences in health literacy, with women being higher in literacy than men and men with low mental health literacy being less likely to recognize and seek care [49,50]. Transgender and gender-diverse individuals experience structural barriers (such as limited accessible competent providers and travel barriers) that will influence how health information is accessed and utilized, even as high literacy results from self-education in some situations [51,52]. Health literacy is not the same across ethnic/racial groups or between migrants, where understanding and application of health information is impaired by language and cultural factors [50,53,54]. In regard to animated media, these disparities indicate that content and delivery should either be health literacy-informed or culturally competent: scripting should be in plain language; if narratives or visuals are used, they should be inclusive and non-stigmatizing; subtitles or voice-overs should be in local languages; descriptive audio and high-contrast/icon-forward formats should be employed where possible; and brief teach-backs or facilitated discussions should also be considered to promote better understanding and action to the extent that they are feasible for animated interventions. Training implementers in cultural competence and health-literacy-sensitive communication may also enhance uptake and equity of outcomes [55]. Future investigations should pre-specify subgroup analyses (e.g., by gender identity/assumed gender, migrant/ethnic status, and literacy levels) and report equity-sensitive outcomes in addition to overall outcomes.

### 4.6. Limitations

This study has several limitations. First, this review was limited to English-language and peer-reviewed articles; the grey literature and unpublished studies were not included. These decisions introduce potential language and publication biases and may result in the omission of relevant studies. However, the use of peer-reviewed articles guarantees that the included evidence was produced at a quality level with some standards. Future reviews might consider expanding the scope to include the non-English and grey literature to reduce selection bias. Second, substantial heterogeneity was observed among populations, sexual health topics, intervention types, comparator groups, assessment tools, and follow-ups. As such, a quantitative meta-analysis was not possible; narrative synthesis was used, and the results should be treated with caution. Furthermore, future research with more standardized and methodologically consistent studies would facilitate meta-analysis. Third, although the search strategy explored both 2D and 3D animation modalities, none of the included studies explicitly mentioned 3D or mixed formats. Thus, it was not possible to assess any possible differences in terms of learning effects across the different types of animation. In addition, although user interactivity and cultural adaptation were taken from the studies and summarized, the reporting was inconsistent, thereby inhibiting the assessment of the independent effects of these variables on the outcomes. Future studies should describe and assess these variables explicitly to enable more systematic comparisons and quantitative synthesis. Fourth, data were almost exclusively self-reported and used instruments of varying levels of validation, with harms or unintended consequences seldom addressed. Well-prepared research would use multiple sources of data and standardized measures in the assessment. Fifth, there was a lack of reporting of intervention fidelity, adherence, production quality, or implementation context (e.g., facilitator training, costs, and scalability); as such, it was difficult to evaluate feasibility and reproducibility in the real world. Future research should include reporting frameworks to document the delivery of interventions. Sixth, evidence was gathered from various geographic locations but primarily clumped in only a few countries, limiting the generalizability and implications for different cultural and educational contexts. Future research conducted in different geographic or sociocultural contexts would enhance the external validity and culturally relevant interventions. Finally, several included articles exhibited a serious or moderate risk of bias, specifically in the group of non-RCTs, which had issues with confounding, participant selection, and outcome measurement. Concerns regarding outcome assessment were noted, even in RCTs. The identified risk of bias could compromise the trustworthiness of the synthesized evidence, so results should be interpreted with caution. Taken together, the predominance of non-randomized designs, serious or moderate risk of bias ratings, and inconsistent outcome measures suggest that the overall strength of evidence in favor of motion graphics and animated videos in sexual health education is low to moderate and should be interpreted cautiously.

Despite the possibilities for animated and motion-graphic interventions as methods of communication, there is currently only a limited evidence base in terms of size, consistency, and rigor of methods. As a result, large-scale implementation or cost-effectiveness cannot be justified at this point. Nevertheless, the findings from this study suggest that these types of media may supplement existing health-education strategies, assuming that future research generates stronger evidence of effectiveness and sustainability. Future studies should evaluate implementation and economic outcomes (e.g., production costs, delivery feasibility, and scale across settings), which would inform public health planning at a more impactful level.

## 5. Conclusions

This systematic review emphasizes the promise of multimedia (motion graphics and animated videos) as a viable approach to health, health promotion, and sexual health messages. Multimedia consistently improved knowledge and attitudes in several audience contexts and, on some occasions, influenced intentions, behaviors, and psychosocial outcomes. Both short, scalable campaigns and more complex interventions that incorporated multiple sessions were effective; however, the authors found sustained change was most likely when the content was recurring and grounded in theory and the social context. Prior research has suggested using theories and frameworks for developing culturally tailored and theory-based interventions, including SCT, HBM, PWM, and IMB. Motion graphics and animated videos also provide a safe and engaging medium to address sensitive topics and serve as an avenue to alleviate stigma and embrace inclusivity in sexual health messages. Future studies should focus on standardization of the design and content, long-term effects, and facilitating implementation (i.e., fidelity, scalability, and cost). In conclusion, animated and motion graphics media have educational potential, but the evidence is insufficient to determine their widespread use or cost-effectiveness. Further research is needed to determine their long-term effects on learning outcomes and best practices for integrating these tools into educational settings. Technology will need to be tested in diverse learning environments as it advances. Motion graphics and animated media show promise for improving sexual-health knowledge and attitudes and, in select contexts, intentions and behaviors. However, the body of evidence is small and heterogeneous, with frequent high risk of bias; certainty is therefore low to moderate. Future research should prioritize standardized outcomes, longer follow-up, and implementation and cost-effectiveness evaluations before advocating large-scale adoption.

## Figures and Tables

**Figure 1 healthcare-13-02895-f001:**
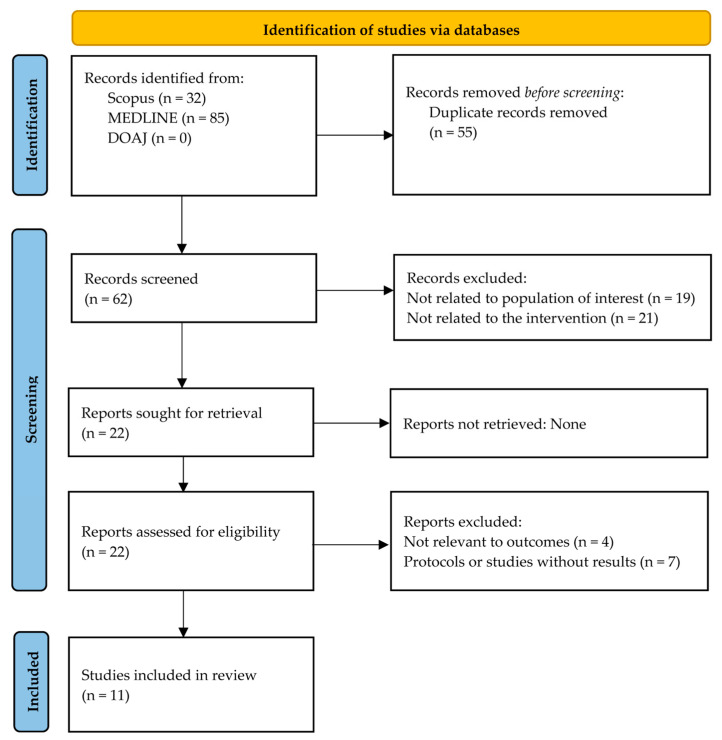
PRISMA flow diagram of the study selection process. DOAJ, Directory of Open Access Journals; MEDLINE, Medical Literature Analysis and Retrieval System Online; PRISMA, Preferred Reporting Items for Systematic Reviews and Meta-Analyses.

**Figure 2 healthcare-13-02895-f002:**
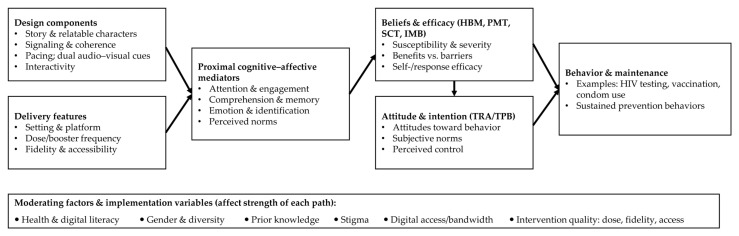
Conceptual synthesis and unified framework for animated/motion-graphic sexual health interventions. HBM, Health Belief Model; PMT, Protection Motivation Theory; TRA, Theory of Reasoned Action; TPB, Theory of Planned Behavior; IMB, Information–Motivation–Behavioral Skills model; SCT, Social Cognitive Theory.

**Table 1 healthcare-13-02895-t001:** Characteristics of included studies.

Authors, Year	Country	Participant Characteristics	Study Design	Intervention Details	Assessment Tools	Main Findings
Li et al., 2025 [26]	China	335 university students (mean age 21.4; 63.6% female)	Cross-sectional survey; pre- and immediate post-intervention	2 min animated video on HPV prevention	- BeSD-based questionnaire - 10 HPV cognition items	- 88% willing to vaccinate; 52% already vaccinated - Willingness associated with female sex, higher expenses, sexual activity, social support - Video effectively improved vaccination willingness
Huang et al., 2024 [25]	China	243 vocational high school students (mean age 17.6; ~50% male)	Cluster RCT; baseline & post-intervention	6-lesson curriculum with short animations, group discussions, quizzes	- ATLG scale - Researcher-made questionnaire - Feedback survey	- Significant improvements in attitudes & knowledge (*p* < 0.001) - Female students had more positive attitudes - 98.5% favored animation-based curricula
Zhou et al., 2024 [29]	China	1835 + 374 (ages 9–12)	Repeated intervention with PSM & DID analysis; post & 1-year follow-up	6-session animation-based sexuality education	- 48 knowledge Qs - 27 attitude items- Binary abuse/pornography awareness	- Immediate: knowledge increased by 3.9 points, attitudes increased by 0.7 points (*p* < 0.01) - 1 year: knowledge gains sustained, attitudes slightly declined - Repeated intervention increased knowledge but no effect on attitudes/behaviors
Mansor et al., 2023 [33]	Malaysia	300 diploma students (aged 18–24); attrition 8.3% (n = 25)	2-arm, parallel, single-blinded cluster RCT; baseline, post, 3-mo follow-up	Sexting Intervention Module (SIM), based on Prototype Willingness Model, delivered via 5 animated YouTube videos; waitlist control	Web-based questionnaires (15-item intention, 12-item willingness, 8-item knowledge, 22-item attitudes, 16-item norms, 9-item prototypes)	Significant improvements vs. control (adjusted for age, sex, relationship status, web time): - Decreased intention to sext (β = −0.12, *p* = 0.002, d = 0.23) - Decreased willingness (β = −0.16, *p* < 0.001, d = 0.40) - Increased knowledge (β = 0.12, *p* < 0.001, d = 0.39) - Attitude more positive (β = −0.11, *p* = 0.001, d = 0.31) - Decreased perceived norms (β = −0.07, *p* = 0.04, d = 0.18) - Prototype perceptions more negative (β = −0.11, *p* < 0.001, d = 0.35)
Wang et al., 2023 [28]	China	1725 students in grades 4–6 (aged 9–13; mean 10.65 years; 54.5% male)	Quasi-experimental study with intervention and control groups (propensity score matching); single post-test at 3 months	Six 45 min sexuality education sessions led by trained schoolteachers using a standardized comprehensive curriculum with cartoon animation	Internet-based questionnaires on knowledge, attitudes, and practices	- Intervention group scored higher in knowledge (+3.35/38) and attitudes (+2.02/34). - Improved genital hygiene practices, with stronger effects among boys. - Curriculum effectively enhanced knowledge, skills, attitudes, and practices.
Holst et al., 2022 [34]	Tanzania	600 community participants from rural villages (aged 15–45 years; mean 30.0 ± 8.3 years; 60.3% female)	Nonrandomized community-based longitudinal study; baseline, 3 months, and 12 months	Three animated health videos (HIV/AIDS, tuberculosis, Taenia solium [neuro]cysticercosis and taeniosis) shown on tablets. After 6 months, free access to community InfoSpots with integrated digital health education.	Open-ended Swahili-based questionnaire (binary correct/incorrect scoring)	- At 12 months, knowledge was higher in intervention vs. control: HIV/AIDS +10.2%, TB +12%, T. solium +31.5%. - Knowledge improved across all domains (transmission, symptoms, treatment, prevention). - Participants who used InfoSpots showed greater knowledge gains than non-users.
Nestadt et al., 2019 [35]	Thailand	88 dyads of perinatally HIV-infected (PHIV) youth (aged 9–14; mean 12.3 ± 1.4; 49% female) and their caregivers	Pilot randomized controlled trial; baseline, 6 months, 9 months (3-mo follow-up)	CHAMP+ Thailand (“Walking Together”)—family-based psychosocial program delivered over 6 months in 11 cartoon-based sessions covering HIV stigma, adherence, disclosure, grief/loss, puberty, social support, and communication	- Strengths and Difficulties Questionnaire (SDQ) - Children’s Depression Inventory (CDI) - HIV knowledge questionnaire - Family communication scale - HIV stigma scale - ART adherence self-report + viral load data	- Significant improvements at 6 months in mental health, adherence, HIV knowledge, communication, stigma, and social support. - Most improvements sustained at 9 months. - CHAMP+ group showed greater gains than standard care. - 100% completion of intervention.
Grieb et al., 2017 [31]	USA	104 foreign-born Latino men (mean age 38.5 ± 12.2; 100% male)	Pilot pre–post evaluation; baseline and 6-month follow-up	Individually tailored animated video modules addressing barriers to HIV testing. Participants viewed one of three modules, selected according to their self-reported reason for not testing.	Survey on HIV risk, barriers to testing, and intention to test in the next 3 months	- 50% had never previously tested for HIV. - Intention to test increased significantly after the module (t = –8.28, *p* < 0.001). - Mean likelihood of testing in 3 months rose from 4.9 (SD 3.6) to 7.0 (SD 3.3). - 97% reported learning something new and found the video interesting.
Mi et al., 2015 [27]	China	202 HIV-positive men who have sex with men (MSM) (mean age 30.9; range 18–52 years)	Quasi-randomized web-based intervention; baseline and 6-month follow-up	Internet-based program with four modules: information exchange website, bulletin board system, individualized online counseling with peer educators, and animation game	Confidential interviewer-administered questionnaires at baseline and 6 months	Significant increases in: - Disclosure of HIV status to partners (76.0% vs. 61.2%, *p* = 0.039) - Motivating partners to test for HIV (42.3% vs. 25.5%, *p* = 0.016) No significant differences in early treatment uptake or consistent condom use.
Brock & Smith, 2007 [30]	United States	51 college students (mean age 42.1; range 25–70 years; 51% male)	Quasi-experimental prospective technology intervention; immediate post-test and 4–6 week follow-up	17 min HIV/AIDS educational digital video delivered via personal digital assistant (PDA)	- Pre- and post-surveys - Rapid Estimate of Adult Literacy in Medicine (REALM) - 9-item validated adherence scale	Significant improvements in: - Disease knowledge (*p* < 0.005) - Medication knowledge (*p* < 0.005) - Adherence behavior knowledge (*p* < 0.05) Improved self-reported adherence at 4–6 weeks (*p* < 0.005).
Paperny & Starn, 1989 [32]	United States	718 high school students (aged 13–18; mean age 15.3; 53.7% female) in a health clinic setting	Comparative design with experimental and control groups; immediate post-test	Computer-assisted instruction games: - The Baby Game (n = 351): 30 min color action game on parenting readiness and costs - Romance (n = 367): 40 min color action game on contraception, sexual decision-making, and attitudes	Pre- and post-test instruments on knowledge and attitudes; demographic and evaluation questions	- The Baby Game: Increased awareness of childbirth/child-rearing costs (χ^2^ up to 63.3, *p* < 0.0001); reduced desire for teen parenthood (from 74% to 81%). - Romance: Improved contraceptive knowledge and acceptance (*p* < 0.01); increased willingness to seek professional advice (*p* < 0.01). - Students rated both games positively. Clinics using them reported a 15% decline in adolescent positive pregnancy tests within 1 year.

Note. ART, antiretroviral therapy; ATLG, Attitudes Toward Lesbians and Gay Men scale; BeSD, Behavioral and Social Drivers of Vac-cination framework; CDI, Children’s Depression Inventory; CHAMP+, Collaborative HIV Prevention and Adolescent Mental Health Pro-gram; DID, difference-in-difference; MSM, men who have sex with men; PDA, personal digital assistant; PHIV, perinatally HIV-infected; PSM, propensity score matching; PWM, Prototype Willingness Model; RCT, randomized controlled trial; REALM, Rapid Estimate of Adult Literacy in Medicine; SD, standard deviation; SDQ, Strengths and Difficulties Questionnaire; SIM, Sexting Intervention Module.

**Table 2 healthcare-13-02895-t002:** Educational and multimedia characteristics of motion graphics and animated video interventions.

Authors, Year	Type of Animation (Modality of Animation)	Duration and Frequency	Content Focus	Delivery Platform	Pedagogical/Educational Design Features	Production and Expertise	User Interactivity	Cultural/Contextual Adaptation
Li et al., 2025 [26]	Short animated science popularization video (NR)	One-time intervention (~2 min)	Personalized sexual health education (HPV infection and prevention)	Digital short video during online survey; viewed on participants’ own devices (mobile/computer)	Based on Behavioral and Social Drivers framework (thinking/feeling, social processes, practical issues, motivation); designed as brief, engaging, and easily disseminated video for young audiences; included pre- and post-video questionnaires	Developed under the guidance of HPV and vaccination experts; based on literature review and expert consultation; created by an academic research team in collaboration with regional public health authorities	NR	NR
Huang et al., 2024 [25]	Animated videos (2D)	Six lessons over 3 weeks; each lesson ~45 min (5–10 min animation + 20 min group discussion + 15 min quiz)	Sexual health knowledge and safe practices	In-person classroom delivery by trained teachers; animation integrated into regular lessons	Based on National Sexuality Education Standards and UNESCO frameworks; culturally tailored with non-judgmental language; combined animations with interactive teaching (discussions, quizzes); addressed sensitive issues (LGBTQ, stigma, HIV prevention) in a structured environment	Curriculum content developed with educational and health guidelines; delivered by trained, experienced teachers	Each lesson comprised a short 2D animated segment, followed by group discussion and an in-class quiz.	Content tailored to Chinese legal & socio-cultural context; neutral/age-appropriate language
Zhou et al., 2024 [29]	Animation-based comprehensive sexuality education package (NR)	Six 45 min sessions delivered over 2 months (September–November 2020); subgroup (n = 374) received a repeat round in September 2021	Sexual health education (reproduction, puberty, body knowledge, respect for others)	In-person classroom sessions in rural schools, integrated into weekday classes	Guided by UNFPA International Technical Guidance on Sexuality Education; informed by social norms theory; package included animated videos, handbooks, scripts, and slides; teachers shown reference video-recordings to standardize delivery; interactive class discussions and Q&A facilitated by teachers	Content reviewed by sexual/reproductive health experts, frontline educators, and students; teachers trained by research staff before implementation	Newsletters and small-group discussion with game/interactive activities	NR
Mansor et al., 2023 [33]	Digital animated videos	5 videos (~24 min total) delivered over 3 consecutive days (NR)	Sexting prevention	Private YouTube link; follow-up discussion encouraged via WhatsApp chat	Based on Prototype Willingness Model (attitudes, perceived norms, prototype perceptions); integrated behavior change techniques (prompting, preventive instruction); audio-visual design simplified complex information to improve retention and engagement; pre-tested among 30 diploma students before implementation	Content validated by public health experts and clinical psychologists; delivered by the primary researcher (physician)	Watched short YouTube animations; optional WhatsApp chat with researchers for Q&A	Framed within Malaysian legal/school context and local issues
Wang et al., 2023 [28]	Cartoon animation series (NR)	Six sessions, each 45 min, delivered once	HIV/STD prevention	In-person classroom sessions, teacher-facilitated using standardized package (scripts, slides, model educator videos)	Based on UNESCO International Technical Guidance on Sexuality Education (2018); teachers received 2-day training (desensitization, package use, trial lessons); included interactive group discussions and tasks; cognitive pre-test ensured questionnaire alignment with students’ comprehension	Developed by sexuality education professionals, teachers, and researchers; standardized to reduce variability in delivery, particularly in rural areas with limited trained educators	NR	NR
Holst et al., 2022 [34]	Animated health videos (NR)	Three short videos (3–7 min each)	Sexuality and health education (HIV/AIDS, tuberculosis, *Taenia solium*)	Tablet-based viewing at home (baseline); after 6 months, community InfoSpots with smartphones/tablets providing open access to animations, text, graphics, and quizzes	Developed with local stakeholders and government-approved health promotion materials; bilingual (Swahili and English); reinforcement through repeated messages with illustrations and narration	Produced by an interdisciplinary team; co-designed with local stakeholders to ensure cultural relevance and acceptability	NR	NR
Nestadt et al., 2019 [35]	Cartoon-based curriculum (2D)	11 sessions over 6 months; delivered one weekend per month, with two sessions per day (except final session)	Sexual violence prevention and family strengthening	In-person sessions at four HIV clinics; group formats included youth-only, caregiver-only, and combined sessions	Grounded in Social Action Theory (SAT); incorporated storytelling and cartoons to reduce literacy barriers; included ice-breakers, group discussions, and follow-up exercises; emphasized family strengthening (communication, problem-solving, peer negotiation skills)	Adapted collaboratively by Thai and U.S. researchers, clinicians, and families using a community-based participatory approach; delivered by trained social workers and counselors following a structured facilitator’s manual	11 cartoon-based sessions; ice-breakers, guided activities, small-group & family discussions each session	CHAMP+ adapted for Thailand; cartoon format used for literacy/sensitive topics; co-developed with Thai stakeholders.
Grieb et al., 2017 [31]	Graphic animation modules (NR)	Three modules, each with three short stories; individual viewing once per participant during survey (10–15 min total)	HIV/STI prevention and HIV testing promotion	Tablet computer, facilitated by research assistants in community and street venues	Guided by the Information–Motivation–Behavioral Skills model and Transtheoretical Model; module tailored to participant’s self-reported testing barrier; included culturally sensitive storytelling (Latino male characters, Spanish language, community-based issues); concluded with local Spanish-language HIV testing site information	Developed by researchers, community leaders, and design students; created through an iterative, community-driven design process with focus groups refining stories, language, and visuals	Viewing graphic-animation modules as part of a campaign	Iterative co-design with Latino immigrants; Spanish branding/name, colors, storylines chosen via 10 focus groups
Mi et al., 2015 [27]	Animation game (NR)	Intervention lasted 6 months	Sex education program (safe sex, disclosure, HIV knowledge, motivation)	Online/web-based program (website, bulletin board, QQ counseling, animated game); accessed with researcher-provided authorization code	Animation game generated personalized feedback letters with tailored safe sex advice based on player’s choices; integrated peer educator counseling with interactive web tools; peer educators posted FAQs and answered questions online; repeated engagement reinforced disclosure, motivation, and HIV knowledge	Developed through collaboration between national/regional public health authorities and a community-based LGBT health organization; delivered by trained HIV-positive MSM peer educators (25 h of training)	Web intervention included an animation game plus peer-counseling component	Designed for Chinese MSM in Chengdu; local delivery with peer counselors
Brock & Smith, 2007 [30]	Audiovisual digital video with animation/graphics (NR)	Single 17 min video; viewed once during clinic visit, with follow-up survey at 4–6 weeks	Sexuality education and HIV treatment adherence	PDA device (handheld, portable) in outpatient infectious disease clinic; viewed in exam rooms, waiting rooms, or labs with headphones for privacy	Grounded in self-efficacy theory; included audio narration of survey questions for participants with low literacy; used realistic patient scenarios and adherence messages; designed for portability and usability in busy clinical environments	Developed by researchers; content adapted from pharmacist-led instruction studies; reviewed by clinicians and patients before finalization	Computer-based video modules accessible via clinic tool; repeated exposure and topic selection (computer-mediated)	NR
Paperny & Starn, 1989 [32]	Animated-action computer games with interactive simulation features (NR)	One-time play; each session 30–40 min during school health class	Pregnancy prevention and sexual decision-making	TRS-80 color computer (64K) in classrooms; implemented in both public and private high schools across socioeconomic groups	Grounded in social learning theory; interactive simulations where students made choices and experienced consequences (pregnancy, costs, parenting demands); included worksheets for reinforcement; emphasized decision-making, communication skills, contraception awareness, and attitude change	Developed with cooperation of ~200 teenagers (pilot tested and revised multiple times); written at sixth-grade reading level for accessibility; supported by pediatricians, gynecologists, and educators	Interactive computer exercises and animated games; simulation-style learning	NR

Note. 2D, two-dimensional; NR, not reported; IMB, Information–Motivation–Behavioral Skills model; MSM, men who have sex with men; PDA, personal digital assistant; PWM, Prototype Willingness Model; SAT, Social Action Theory; STI, sexually transmitted infection; UNFPA, United Nations Population Fund; UNESCO, United Nations Educational, Scientific and Cultural Organization.

**Table 3 healthcare-13-02895-t003:** Risk of bias assessment for randomized controlled trials using Cochrane Risk of Bias 2 tool (RoB2).

Author (Year)	D1	D2	D3	D4	D5	Overall
Huang et al., 2024 [25]	Low risk	Low risk	Low risk	Some concerns	Low risk	Some concerns
Mansor et al., 2023 [33]	Low risk	Low risk	Low risk	Some concerns	Low risk	Some concerns
Nestadt et al., 2019 [35]	Low risk	Low risk	Low risk	Some concerns	Low risk	Some concerns

Note. D1, Bias arising from the randomization process; D2, Bias due to deviations from intended interventions; D3, Bias due to missing outcome data; D4, Bias in measurement of the outcome; D5, Bias in selection of the reported result.

**Table 4 healthcare-13-02895-t004:** Risk of bias assessment for non-randomized studies of interventions using Risk Of Bias In Non-randomized Studies of Interventions (ROBINS-I).

Author (Year)	D1	D2	D3	D4	D5	D6	D7	Overall
Li et al., 2025 [26]	Serious risk	Moderate risk	Low risk	Low risk	Low risk	Serious risk	Moderate risk	Serious risk
Zhou et al., 2024 [29]	Serious risk	Moderate risk	Low risk	Low risk	Moderate risk	Serious risk	Low risk	Serious risk
Wang et al., 2023 [28]	Serious risk	Moderate risk	Low risk	Low risk	Moderate risk	Serious risk	Low risk	Serious risk
Holst et al., 2022 [34]	Serious risk	Moderate risk	Low risk	Low risk	Moderate risk	Moderate risk	Low risk	Serious risk
Grieb et al., 2017 [31]	Serious risk	Serious risk	Low risk	Low risk	Low risk	Serious risk	Moderate risk	Serious risk
Mi et al., 2015 [27]	Moderate risk	Moderate risk	Low risk	Low risk	Low risk	Serious risk	Low risk	Serious risk
Brock & Smith, 2007 [30]	Serious risk	Moderate risk	Low risk	Low risk	Moderate risk	Serious risk	Moderate risk	Serious risk
Paperny & Starn, 1989 [32]	Serious risk	Moderate risk	Low risk	Low risk	Low risk	Serious risk	Moderate risk	Serious risk

Note. D1, Bias due to confounding; D2, Bias in selection of participants; D3, Bias in classification of interventions; D4, Bias due to deviations from intended interventions; D5, Bias due to missing data; D6, Bias in measurement of outcomes; D7, Bias in selection of the reported result.

**Table 5 healthcare-13-02895-t005:** Summary of findings across included studies for knowledge gain, attitude change, behavioral intention, and psychosocial well-being.

Study	Design/N	Knowledge Gain	Attitude Change	Behavioral Intention	Psychosocial Well-Being	Key Effect Details	Follow-Up
Li et al., 2025 (China) [26]	Cross-sectional pre–post; n = 335	NR (cognition items collected)	NR	↑ Willingness to vaccinate after 2 min video	NR	88% willing; 52% already vaccinated; willingness ↑ post-video (no effect size reported)	Immediate post intervention
Huang et al., 2024 (China) [25]	Cluster RCT; n = 243	↑ Significant (*p* < 0.001)	↑ Significant (*p* < 0.001)	NR	NR	Female students had more positive attitudes; 98.5% favored animation-based curriculum	Post-intervention
Zhou et al., 2024 (China) [29]	Repeated program with PSM & DID; n = 1835 + 374	↑ +3.9 points immediate; sustained at 1 y (*p* < 0.01)	↑ +0.7 immediate; slight decline at 1 y	↔ No effect reported on behaviors/intent	NR	Repeat round ↑ knowledge; no effect on attitudes/behaviors beyond immediate	Immediate & 1-year
Mansor et al., 2023 (Malaysia) [33]	Cluster RCT; n = 300	↑ β = 0.12, *p* < 0.001 (d ≈ 0.39)	↑ β = −0.11, *p* = 0.001 (d ≈ 0.31)	↓ Intention (β = −0.12, *p* = 0.002, d ≈ 0.23); ↓ Willingness (β = −0.16, *p* < 0.001, d ≈ 0.40)	NR	Norms ↓ (β = −0.07, *p* = 0.04); Prototypes more negative (β = −0.11, *p* < 0.001)	Post & 3 months
Wang et al., 2023 (China) [28]	Quasi-experimental (PSM); n = 1725	↑ +3.35/38	↑ +2.02/34	↑ Improved hygiene practices	NR	Enhanced knowledge, skills, attitudes, practices vs. control	3 months post
Holst et al., 2022 (Tanzania) [34]	Nonrandomized longitudinal; n = 600	↑ HIV +10.2%, TB +12%, T. solium +31.5% at 12 m	NR	NR	NR	InfoSpot users showed greater gains	3 & 12 months
Nestadt et al., 2019 (Thailand) [35]	Pilot RCT; 88 dyads	↑ HIV knowledge	↑ Stigma ↓; communication ↑	↑ ART adherence & related behaviors	↑ Mental health, social support	Gains at 6 months; most sustained at 9 months; 100% completion	6 & 9 months
Grieb et al., 2017 (USA) [31]	Pre–post; n = 104	↑ (self-reported learning)	NR	↑ Intention to test (t = −8.28, *p* < 0.001)	NR	Mean likelihood to test from 4.9 to 7.0/10; 97% found video interesting	Immediate & 6 months
Mi et al., 2015 (China) [27]	Quasi-randomized web; n = 202 MSM (HIV+)	NR	NR	↑ Disclosure to partners from 61.2% to 76.0% (*p* = 0.039); ↑ Motivating partners to test from 25.5% to 42.3% (*p* = 0.016)	NR	No differences in early treatment uptake or consistent condom use	6 months
Brock & Smith, 2007 (USA) [30]	Quasi-experimental; n = 51	↑ Disease & medication knowledge (*p* < 0.005)	NR	↑ Self-reported adherence at 4–6 weeks (*p* < 0.005)	NR	Adherence behavior knowledge ↑ (*p* < 0.05)	Immediate & 4–6 weeks
Paperny & Starn, 1989 (USA) [32]	Comparative classroom; n = 718	↑ Contraceptive knowledge (*p* < 0.01)	↑ Acceptance of contraception (*p* < 0.01)	↑ Willingness to seek professional advice (*p* < 0.01)	NR	Clinics reported 15% decline in positive pregnancy tests within 1 year	Immediate (clinic trend at 1 year)

Note. Direction indicates change compared with baseline or comparator. NR = not reported; ↑ = improvement; ↔ = no meaningful change; ↓ = worsening; β = standardized regression coefficient; Δ = mean difference; PSM = propensity score matching; DID = difference-in-differences; RCT = randomized controlled trial; MSM = men who have sex with men; ART = antiretroviral therapy; HIV = human immunodeficiency virus. Effect sizes (SMD, d, β, Δ) and *p*-values are reported when available.

## Data Availability

All data supporting the reported results are included in this article.

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
