# Peer review of "Digital Motion Graphics and Animated Media in Health Communication: A Systematic Review of Strategies for Sexual Health Messaging"

_healthcare, 2025, doi:10.3390/healthcare13222895_

Round 1
Reviewer 1 Report
Comments and Suggestions for Authors
Title: Digital Motion Graphics and Animated Media in Health Communication: A Systematic Review of Strategies for Sexual Health Messaging
Dear authors, thank you for the opportunity to review your work. I believe I have little to add methodologically to improve it. It is a well-executed piece of work. The topic is original and interesting. The writing is clear and easy to read. I reiterate my gratitude.
- Is it possible that the authors could have continued the identification process until they obtained a more relevant number of documents? This is a complex decision for a reviewer. It implies a substantial modification of the work. In a way, it means starting the work all over again.
- Along the same lines, could I say that the number of documents they ultimately used to answer their research objectives is very small, only eleven? I encounter the same problem.
This is the weak point of the work. On the other hand, it is a methodologically sound piece of work.
JUSTIFICATION FOR THE REVISION PROPOSAL:
Methodologically, I have little to contribute to its improvement. It is a well-done piece of work.
The Prisma checklist has been followed scrupulously.
The topic is original and interesting.
The writing is clear and easy to read.
1. Title: The authors indicate that this is a systematic review
2. Research objective: The research objective is well expressed and it is clear what the authors intend. “This systematic review examines the use of motion graphics and animated videos in sexual health education. More specifically, the review will (i) describe the features and design characteristics of these interventions; (ii) describe the educational and behavioral outcomes addressed with motion graphics and animated videos in sexual health education; and (iii) summarize the evidence of the effectiveness of motion graphics and animated videos in improving knowledge, attitudes, behaviors, and related outcomes in sexual health”.
3. Materials and methods
a) The authors specify that they have registered the review protocol.
b) They clearly indicate the eligibility criteria and the sources of information. They include the search algorithm in a practical way.
c) They identify the selection procedure.
d) The review report allows for the replication of the process followed by the authors in the construction of the work. This is an element that is not very common and which I consider relevant.
e) The authors assessed bias using the Cochrane Risk tool, with two reviewers working independently. Disagreements were resolved through discussion, and a third reviewer was consulted when necessary.
4. Results
The presentation of the results is clear and comprehensive. The description of the different studies is very appropriate.
The organization of the results by headings is very effective.
5. Discussion
The discussion of the results is easy to read and addresses the issues in an organized manner. The discussion is well-supported by the results.
Author Response
Dear Reviewer 1,
Title: Digital Motion Graphics and Animated Media in Health Communication: A Systematic Review of Strategies for Sexual Health Messaging
Comment: Dear authors, thank you for the opportunity to review your work. I believe I have little to add methodologically to improve it. It is a well-executed piece of work. The topic is original and interesting. The writing is clear and easy to read. I reiterate my gratitude.
- Is it possible that the authors could have continued the identification process until they obtained a more relevant number of documents? This is a complex decision for a reviewer. It implies a substantial modification of the work. In a way, it means starting the work all over again.
- Along the same lines, could I say that the number of documents they ultimately used to answer their research objectives is very small, only eleven? I encounter the same problem.
This is the weak point of the work. On the other hand, it is a methodologically sound piece of work.
Reply
We appreciate your thoughtful comments regarding the topic of yield. Our research design followed a pre-registered protocol detailing databases, eligibility criteria, and the timeframe for the investigation to attempt to reduce bias and promote generalizability and reproducibility. To attempt to expand the identification of studies post hoc (e.g., adding specific types of studies or broadening the definition of criteria) would undermine our a priori protocol and introduce additional bias. The small number of studies (n = 11) reflects the current state of empirical research in the area of motion graphics and animated interventions for sexual health outcomes rather than an incomplete search process. We have clarified our screening processes and protocol-defined area of scope and enhanced our Limitations and Implications Discussion sections to reinforce the idea that it is the modest evidence base that limits the strength and generalizability of conclusions and that methodological integrity is maintained.
JUSTIFICATION FOR THE REVISION PROPOSAL:
Methodologically, I have little to contribute to its improvement. It is a well-done piece of work.
The Prisma checklist has been followed scrupulously.
The topic is original and interesting.
The writing is clear and easy to read.
- Title:The authors indicate that this is a systematic review
Reply
We are grateful for the positive evaluation and comments regarding the methodological rigor, clarity, and originality of our work. Furthermore, we were enthused to have our adherence to the PRISMA checklist acknowledged and that the title of our manuscript accurately reflects the design of our systematic review.
- Research objective:The research objective is well expressed and it is clear what the authors intend. “This systematic review examines the use of motion graphics and animated videos in sexual health education. More specifically, the review will (i) describe the features and design characteristics of these interventions; (ii) describe the educational and behavioral outcomes addressed with motion graphics and animated videos in sexual health education; and (iii) summarize the evidence of the effectiveness of motion graphics and animated videos in improving knowledge, attitudes, behaviors, and related outcomes in sexual health”.
Reply
We sincerely thank the reviewer for your recognition of the clarity and appropriateness of our research objectives. We appreciate the positive comments on the objectives of our review, and we were pleased to hear they were conveyed appropriately and aligned with our study focus.
- Materials and methods
- a) The authors specify that they have registered the review protocol.
- b) They clearly indicate the eligibility criteria and the sources of information. They include the search algorithm in a practical way.
- c) They identify the selection procedure.
- d) The review report allows for the replication of the process followed by the authors in the construction of the work. This is an element that is not very common and which I consider relevant.
- e) The authors assessed bias using the Cochrane Risk tool, with two reviewers working independently. Disagreements were resolved through discussion, and a third reviewer was consulted when necessary.
Reply
We sincerely thank the reviewer for your thoroughness and recognition of the clarity and transparency of our Methods section. We sincerely appreciate your consideration of our protocol registration, eligibility criteria, explicit selection procedures, and independent risk-of-bias assessment, all to provide methodological rigor and reproducibility.
- Results
The presentation of the results is clear and comprehensive. The description of the different studies is very appropriate.
The organization of the results by headings is very effective.
Reply
Thank you for your positive comment and appreciation of our Results section's clarity and organizational and comprehensive nature. We're pleased that the results and structure effectively conveyed the findings from the included studies.
- Discussion
The discussion of the results is easy to read and addresses the issues in an organized manner. The discussion is well-supported by the results
Reply
We sincerely thank the reviewer for your positive feedback and recognition of the clarity and organization of our Discussion sections, and we're pleased to have your recognition of the strong alignment of the Discussion with the Results. Thank you for your positive comments about the readability and clarity in this section.
Reviewer 2 Report
Comments and Suggestions for Authors
The article focuses on a current and innovative topic in health communication and digital media. Given the limited number of systematic reviews on the use of animation and motion graphics in sexual health education, the study makes an original contribution to the literature.
It can be accepted after some revisions.
1) Three databases were searched. Why were additional databases such as Embase, PsycINFO, or Web of Science not included? Inclusion is recommended. Including these databases could increase the number of articles included in the study.
2) The provided search macro includes 2D and 3D options. Factors such as differences in 2D and 3D animation, levels of cultural adaptation, and user interaction levels should be included in the article.
3) For the 11 publications in the study, adding a SoF graph containing Knowledge gain, Attitude change, Behavioral intention, and Psychosocial well-being would improve the readability of the article.
Author Response
Dear Reviewer 2,
Comment: The article focuses on a current and innovative topic in health communication and digital media. Given the limited number of systematic reviews on the use of animation and motion graphics in sexual health education, the study makes an original contribution to the literature.
It can be accepted after some revisions.
1) Three databases were searched. Why were additional databases such as Embase, PsycINFO, or Web of Science not included? Inclusion is recommended. Including these databases could increase the number of articles included in the study.
Reply
We appreciate the reviewer acknowledging the novelty and significance of our work. With respect to the search strategy, we followed a registered a priori protocol for our review that prespecified the three databases we searched in order to maintain methodological consistency and reproducibility of the review process. The specific databases we selected were chosen because each database covers the major sources of health communication, digital media, and behavioral research. If we had expanded our review to other databases such as Embase, PsycINFO, or Web of Science after we registered our protocol, this would be considered a post hoc change and potentially introduce selection bias.
2) The provided search macro includes 2D and 3D options. Factors such as differences in 2D and 3D animation, levels of cultural adaptation, and user interaction levels should be included in the article.
Reply
We are grateful to the reviewer for this important suggestion, which we have answered by extracting and tabulating these factors in Table 2 (“Educational and multimedia factors”), adding the mode of animation (2D/3D/mixed/NR), interactivity, and cultural/contextual adaptation. These elements are also described in the Results section (3.1), where they are also supported with accurate numbers and references. We have added a new paragraph in the Limitations section admitting that none of the included studies had explicitly used 3D or mixed-modality animation and that the reporting of interactivity and cultural adaptation was inconsistent.
3) For the 11 publications in the study, adding a SoF graph containing Knowledge gain, Attitude change, Behavioral intention, and Psychosocial well-being would improve the readability of the article.
Reply
We appreciate the reviewer’s helpful suggestion. A Summary of Findings (SoF) table summarizing knowledge gain, attitude change, behavioral intention, and psychosocial well-being across the 11 included studies has been added (Table 5) to enhance clarity and readability of the results.
Reviewer 3 Report
Comments and Suggestions for Authors
The topic is highly relevant, addressing an innovative intersection between digital media and sexual health promotion. The manuscript explores an emerging area with strong public health implications. The following comments should be addressed before publication:
1) The review follows PRISMA guidelines, but the methodology lacks sufficient detail on data extraction, quality appraisal, and synthesis. Please specify how the strength of evidence was assessed (e.g., GRADE) and clarify how heterogeneity among diverse designs and outcomes was managed, since I cannot see this information in this manuscript.
2) Most included studies present a “serious” risk of bias, yet this is not critically discussed. Can you explain how this limitation affects the overall confidence in your findings and whether any weighting or sensitivity analysis was performed?
3) Several behavioral theories are cited, but without a clear conceptual synthesis. Why? Consider developing a unified framework or diagram illustrating how motion graphics and animations drive behavior change across studies. Please, include this framework since it will be mandatory to address this point.
3) Although cultural adaptation is mentioned, there is limited critical discussion about gender, diversity, and health literacy differences across populations. Reflect on how these factors influence the effectiveness and equity of animated health interventions.
4) Consider adding recent evidence that links health literacy to individuals’ capacity to act on health information (e.g., PMID: 40333082), to highlight that the impact of animated media in sexual health education may depend on users’ health and digital literacy levels.
5) The conclusions remain generic (.....“promising tools for communication”). Strengthen the discussion by outlining real-world implications, barriers to implementation, and cost-effectiveness considerations for large-scale adoption in public health. Expanding on these aspects would better convey the practical significance and policy relevance of your findings.
6) Please revise the abstract to eliminate repetitive content and make it more concise and effective for readers. Also, ensure that references follow MDPI style, maintaining an appropriate balance with about 90% of sources from the last five years and no more than 10% older references.
Author Response
Dear Reviewer 3,
Comment: The topic is highly relevant, addressing an innovative intersection between digital media and sexual health promotion. The manuscript explores an emerging area with strong public health implications. The following comments should be addressed before publication:
1) The review follows PRISMA guidelines, but the methodology lacks sufficient detail on data extraction, quality appraisal, and synthesis. Please specify how the strength of evidence was assessed (e.g., GRADE) and clarify how heterogeneity among diverse designs and outcomes was managed, since I cannot see this information in this manuscript.
Reply
We appreciate the reviewer's insightful comment. We have provided additional information in the Methods section on data extraction, quality assessment, and narrative synthesis. We have included a new subsection on grading the strength of evidence. While we did not formally assess GRADE due to the heterogeneity and limited number of studies included, we did critically assess the strength of evidence through consideration of study design, risk-of-bias ratings, consistency of findings, indirectness, and precision qualitatively. We also clarified how we handled heterogeneity across study designs and outcomes via narrative synthesis and domain-based grouping. Sections 2.5–2.7 of the revised manuscript contain these revisions.
2) Most included studies present a “serious” risk of bias, yet this is not critically discussed. Can you explain how this limitation affects the overall confidence in your findings and whether any weighting or sensitivity analysis was performed?
Reply
We appreciate this beneficial comment. Thank you to the reviewer for giving attention to the methodological quality of the included studies. We have addressed this point in the Discussion (end of Section 4.1), where we include a critical explanatory musing to understand how those identified risks of bias impact our overall confidence in the findings (or lack thereof). Additionally, we discussed that we did not conduct a weighted or sensitivity analysis, again to account for the very high heterogeneity across study designs and outcome measures.
3) Several behavioral theories are cited, but without a clear conceptual synthesis. Why? Consider developing a unified framework or diagram illustrating how motion graphics and animations drive behavior change across studies. Please, include this framework since it will be mandatory to address this point.
Reply
We appreciate the suggestion—thank you! We agree with the reviewer about the need for a clearer conceptual synthesis in the manuscript. Throughout the revisions, the authors created and included a new figure (located in Figure 2) that illustrates a synthesized framework that brings together the behaviorism learning theories cited in the studies presented. The synthesized framework illustrates potential pathways and processes through which the motion-graphic and animated interventions might impact cognitive, affective, and behavioral outcomes.
3) Although cultural adaptation is mentioned, there is limited critical discussion about gender, diversity, and health literacy differences across populations. Reflect on how these factors influence the effectiveness and equity of animated health interventions.
Reply
Thank you for your important, informative feedback. We have added a new subsection in the Discussion titled “Equity considerations: gender, diversity, and health/digital literacy,” which discusses how these factors may influence the effectiveness and equity of animated health interventions. The updated text explains how health literacy varies by gender and diversity, highlights key challenges for transgender and immigrant/migrant groups, and suggests ways to overcome these challenges through culturally aware health literacy design to achieve fairer results.
4) Consider adding recent evidence that links health literacy to individuals’ capacity to act on health information (e.g., PMID: 40333082), to highlight that the impact of animated media in sexual health education may depend on users’ health and digital literacy levels.
Reply
We are grateful for this insightful suggestion. The Discussion section now contains a new subheading recognizing health and digital literacy as possible moderators of the intervention effect, with reference to recently published evidence (Cocchieri et al., 2025). This amendment broadens the implication of health literacy in relation to the study findings while respecting the intended focus of the manuscript.
5) The conclusions remain generic (.....“promising tools for communication”). Strengthen the discussion by outlining real-world implications, barriers to implementation, and cost-effectiveness considerations for large-scale adoption in public health. Expanding on these aspects would better convey the practical significance and policy relevance of your findings.
Reply
We appreciate this valuable suggestion. The Discussion section has been revised to provide balanced reflection on the real-world applicability of the use of animated and motion-graphic interventions given the limited evidence base of what we can generalize to show large-scale implementation in practice. The revised conclusion also highlights other areas of needed future research about long-term effects of animated and motion-graphic interventions, feasibility of implementation in different contexts, and cost-effectiveness of implementation research to increase the practical and policy relevance of future studies.
6) Please revise the abstract to eliminate repetitive content and make it more concise and effective for readers. Also, ensure that references follow MDPI style, maintaining an appropriate balance with about 90% of sources from the last five years and no more than 10% older references.
Reply
Thank you for your feedback. We revised the abstract for conciseness and clarity and adjust the references to meet MDPI style and recency requirements.
Round 2
Reviewer 2 Report
Comments and Suggestions for Authors
Corrections and clarifications made by the author are acceptable.
Reviewer 3 Report
Comments and Suggestions for Authors
Congratulations for the improvement.